# GLOBAL PIVOTS, LOCAL UNKNOWNS: STABLE FEDERATED OPEN-SET SEMI-SUPERVISED LEARNING

## ABSTRACT

We introduce *Federated Open-Set Semi-Supervised Learning (FOSSL)*, a new and practically important federated learning setting where the server holds a small labeled set of in-distribution (ID) classes while clients provide only unlabeled, non-IID data that may include unknown classes. This setting is under-explored and presents two key challenges: pseudo-label brittleness under distributed OOD contamination and amplified heterogeneity arising from diverse OOD categories across clients. These challenges cause conventional federated SSL or centralized OSSL pipelines to become unstable when applied directly. We propose *OpenFL*, a server-guided framework designed to remain robust under these FOSSL-specific difficulties. *OpenFL* stabilizes global training via a round-wise EMA model, maintains class-level pivots as global anchors for representation learning, and aggregates clients using reliability-aware weights. Clients perform gated pivot alignment, strengthening ID-consistent updates while suppressing the influence of uncertain or OOD-prone samples. Across CIFAR-10, CIFAR-100, and Fashion-MNIST with diverse inlier/outlier splits and unseen OOD tests, *OpenFL* improves both ID accuracy and OOD detection while maintaining stable training. This work establishes FOSSL as a benchmark problem and provides a principled framework for learning under unlabeled, open-set, and highly heterogeneous federated environments.

## 1 INTRODUCTION

Modern applications—from camera apps and wearables to home sensors and vehicles—produce privacy-sensitive data that cannot be centralized due to regulation, bandwidth, and trust (European Union, 2016; State of California, 2018). Federated learning (FL) trains where data reside and shares only model updates (McMahan et al., 2017; Bonawitz et al., 2019; Kairouz et al., 2021). In practice, however, clients **rarely curate labels**; at best, a provider maintains a small labeled set on the server while users contribute raw, Non-IID, **open-set** streams mixing in-distribution (ID) examples with unknown out-of-distribution (OOD) classes (Yang et al., 2024).

We study this practically common yet under-explored regime: federated open-set semi-supervised learning (FOSSL) with **labels only at the server**. The server holds a small labeled ID set; each client holds only unlabeled, Non-IID data that may include unknowns. The goal is to learn (1) an accurate ID classifier and (2) a reliable OOD detector *without any client-side labels*. To our knowledge, this *labels-at-server* FOSSL setting has not been systematically formulated and evaluated; see Fig. 1.

Recent work has examined a narrower case, labels-at-server federated semi-supervised learning (FSSL) (Zhang et al., 2021; Diao et al., 2022; Lee et al., 2024), where client unlabeled data are *closed-set* (ID-only). SemiFL (Diao et al., 2022) is a strong template: the server issues client pseudo-labels and, after aggregation, fine-tunes on its small labeled set. Introducing OOD samples fundamentally changes the problem, yielding FOSSL with two key challenges: **(i) Amplified pseudo-label brittleness.** Unseen classes are mapped to ID categories with high confidence, causing systematic mislabeling, class imbalance, and confirmation bias that can lead to collapse. **(ii) Intensified heterogeneity.** Clients face disjoint unknowns, worsening Non-IID drift. This corrupts client-side BN (making server-only statistics indispensable), destabilizes centralized OSSL objectives such as $(K{+}1)$ classification (SCOMatch (Wang et al., 2024)) and subspace-based scoring (ProSub (Wallin et al., 2024)), and induces large round-to-round oscillations, especially when OOD-heavy clients dominate.

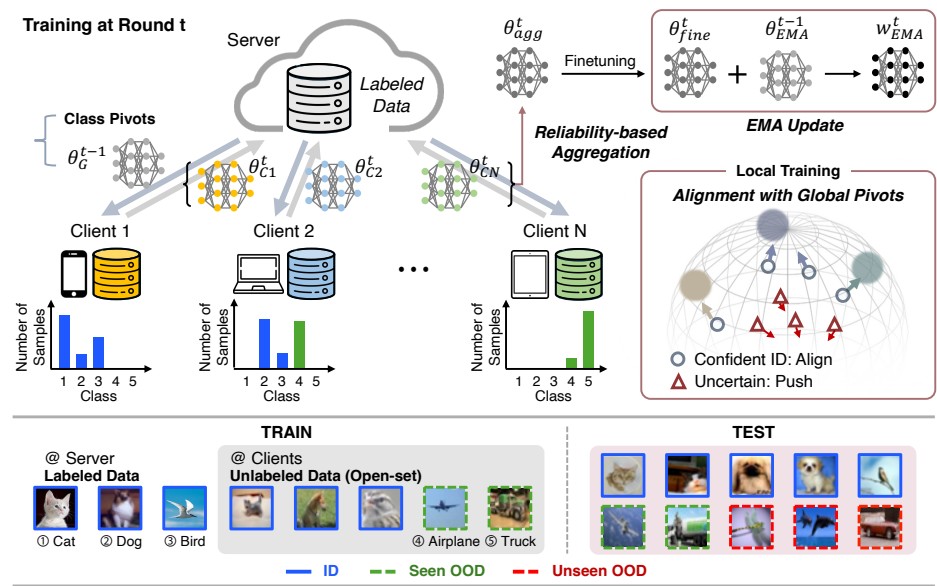

Figure 1: **Overview of the FOSSL problem setting and our method, *OpenFL*.**

Our organizing principle is **Global Pivots, Local Unknowns**: stable, server-anchored references guide learning, while diverse local unknowns are exploited rather than discarded. Through an empirical study, we confirm that server-side fine-tuning (as in SemiFL), server-only BN statistics, and disjoint learning with a one-vs-all (OVA) head form a practical foundation for labels-at-server FOSSL. Building on this, we propose *OpenFL* with three synergistic components:

- **Round-wise EMA (R-EMA):** The server maintains an exponential moving average of the fine-tuned model per *communication round*, smoothing trajectories and mitigating catastrophic forgetting. EMA is not broadcast; it underpins *pivot generation*. This stabilizes pseudo-label brittleness across rounds.

- **Pivot-guided Open-set Alignment:** Using EMA-derived features, the server computes ID pivots (class references). On clients, only *dual-gated, high-confidence* ID samples are attracted toward their pivots, while uncertain/OOD samples receive a mild angular push via the normalization term, improving ID/OOD separation and reducing inter-client divergence. It mitigates pseudo-label brittleness and OOD-induced heterogeneity by selectively.

- **Reliability-Aware Aggregation (RAA):** Clients are weighted by their *alignment quality* (inverse average alignment loss), prioritizing cleaner signals over sheer quantity. Training begins with a short warm-up (consistency only) before enabling alignment. This suppresses unstable updates from clients dominated by OOD samples.

Our contributions are as follows:

- We formalize and systematically study the labels-at-server FOSSL regime, identifying two core challenges: amplified pseudo-label brittleness and intensified heterogeneity from diverse OOD categories.

- We propose *OpenFL*, a server-guided framework that unifies round-wise EMA smoothing, dual-gated pivot alignment, and reliability-aware aggregation into a coherent solution with low communication overhead (EMA remains server-side).

- Through extensive evaluation on CIFAR-10, CIFAR-100, and FashionMNIST with diverse inlier/outlier splits and unseen-OOD tests, *OpenFL* achieves strong inlier accuracy and AUROC with stable convergence, setting robust baselines for this new regime.

## 2 RELATED WORK

**Federated learning (FL).** FL trains models without sharing raw data; FedAvg (McMahan et al., 2017) averages client models by data size. Variants such as FedProx (Li et al., 2020) and FedOpt (Reddi et al., 2021) mitigate Non-IID effects and accelerate convergence.

**Federated semi-supervised learning (FSSL).** FSSL considers clients with limited labels, commonly in two regimes: (i) some labels per client, and (ii) *labels-at-server*, where labeled and unlabeled data are disjoint (the harder case). Early labels-at-server methods—FedMatch (Jeong et al., 2021), FedRGD (Zhang et al., 2021)—combine FedAvg with FixMatch-style losses (Sohn et al., 2020). SemiFL (Diao et al., 2022) provides a strong template: global pseudo-labeling, server fine-tuning on a small labeled set, client static BN (Diao et al., 2021), and FedOpt. (FL)$^2$ (Lee et al., 2024) refines selection via stage-aware thresholds and sharpness-aware consistency, still following this pattern, but still follows the SemiFL training pattern.

**From FSSL to federated open-set semi-supervised learning (FOSSL).** In practice, unlabeled client streams are *open-set*; unknown OOD classes appear during training. This motivates *FOSSL in the labels-at-server setting*, where an ID classifier and an OOD detector must be learned without any client-side labels. To our knowledge, this regime has not been systematically studied. The closest work, FedoSSL (Zhang et al., 2023), targets an open-world variant but *requires client labels*, making it inapplicable here.

**Centralized open-set semi-supervised learning (OSSL).** Applying centralized OSSL to FL is non-trivial. *SCOMatch* (Wang et al., 2024) uses a $(K+1)$ formulation that collapses OOD into one class; under heterogeneous, Non-IID OOD in FL, this superclass can be inconsistently formed across clients. *ProSub* (Wallin et al., 2024) estimates OOD via prototype subspaces and Beta distribution fits, which become unstable with label–unlabeled disjointness and client heterogeneity. One-vs-all (OVA)-based methods—*OpenMatch* (Saito et al., 2021), *IOMatch* (Li et al., 2023), *SSB* (Fan et al., 2023)—use labeled ID-vs-rest heads and are more compatible with labels-at-server, yet their unlabeled objectives (entropy minimization and soft-consistency regularization) tend to be brittle in the *labels-at-server* regime, where clients train solely on unlabeled, open-set, Non-IID streams. To our knowledge, no prior work systematically extends OSSL to labels-at-server FOSSL, underscoring the need for methods robust to intensified heterogeneity and strict label–unlabeled separation.

# 3 EMPIRICAL STUDY OF FOSSL CHALLENGES

Since no prior study has systematically examined FOSSL, we conduct the first foundational study of its unique difficulties by adapting existing methods to this setting. We adapt representative labels-at-server FSSL methods (SemiFL, FedFixMatch) and extend recent OSSL techniques into their federated forms (FedSCOMatch, FedProSub, and FedSSB), with full implementation details in Appendix A.1. Experiments are conducted on CIFAR-10, designating its six animal classes as ID and the remaining four as OOD. The server holds 40 labeled samples per ID class, while 100 clients each contain only unlabeled data mixing ID and OOD samples. Our findings reveal a cascade of failures, motivating the core principles of our proposed method.

**Existing FSSL Degrades Under OOD Contamination.** Standard FSSL frameworks may collapse when exposed to OOD data. SemiFL, for instance, collapses before convergence (Figure 2, top-left) because OOD samples are confidently mislabeled as ID classes, polluting client-side training. This process *amplifies pseudo-label brittleness*, creating severe class imbalance and confirmation bias that rapidly degenerates the model into predicting a single dominant class. While FedFixMatch avoids total collapse (Figure 2, top-right), its OOD discrimination is nullified, rendering it ineffective for the open-set problem.

**StaticBN Becomes an Indispensable Prerequisite in FOSSL.** A key struggle in FOSSL is that client-side OOD data fatally corrupts batch normalization statistics. Our FedFixMatch experiments (Figure 2, top-right) confirm this: performance drops significantly in the open-set scenario and degrades further without server-only statistics ("Open-set w/o SBN"). Since alternatives like local BN or FedBN (Li et al., 2021) are also vulnerable, StaticBN (Diao et al., 2021)—which uses only the server's clean data—is an *essential prerequisite* for stable training. We therefore build all subsequent FOSSL experiments in our study upon StaticBN.

**Disjoint Labels and OOD Heterogeneity Undermine Centralized OSSL Assumptions.** While OSSL methods like SCOMatch and ProSub excel centrally, their federated adaptations fail catastrophically. This collapse is visually stark in the t-SNE (Van der Maaten & Hinton, 2008) plots (Figure 2, bottom), revealing a complete breakdown of the feature space structure. The reasons are fundamental to the FOSSL setting. SCOMatch, which treats all OOD data as a single unified class, breaks down under heterogeneous OOD distributions across clients during aggregation. ProSub's

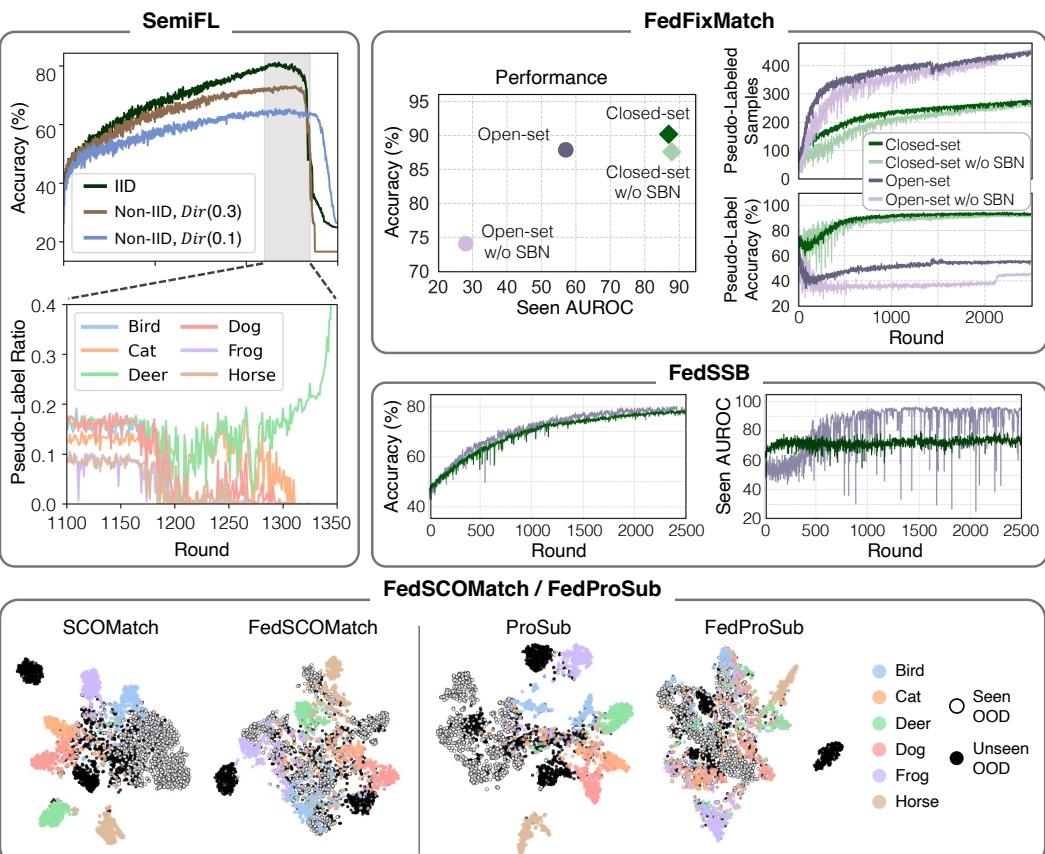

Figure 2: **Existing FSSL and OSSL Methods Fail in the FOSSL Setting.** We adapt representative FSSL methods (SemiFL, FedFixMatch) and federated OSSL variants (FedSSB, FedSCOMatch, and FedProSub) to the labels-at-server FOSSL environment.

reliance on constructing clean ID subspaces and fitting distributions for OOD scoring becomes unreliable when the server holds no OOD examples and client pseudo-labels are noisy. *Ultimately, these methods are not robust to the disjoint-label nature and intensified heterogeneity of FOSSL.*

**OOD Data is a Double-Edged Sword: Useful but Destabilizing.** Interestingly, exposing the model to OOD data is not entirely negative. As suggested by FedSSB's one-vs-all approach—which aligns well with the disjoint setting—leveraging OOD samples during training can improve OOD detection. In FOSSL, however, this benefit is offset by severe instability. Heterogeneous OOD across clients amplifies training fluctuations (Figure 2, middle-right AUROC plot) and leads to catastrophic forgetting of ID representations, particularly in rounds dominated by OOD-heavy clients. *Therefore, a successful FOSSL method must not discard OOD data, but rather incorporate a mechanism to harness its benefits while mitigating the instability it introduces.*

## 4 METHOD

Our method addresses the core challenges of FOSSL by establishing a stable, server-guided learning framework. Building upon the server-client training structure from SemiFL (Diao et al., 2022), we introduce three key components that work in synergy: **(1) a round-wise EMA model** to provide a stable geometric backbone, **(2) pivot-guided alignment** to safely leverage unlabeled client data, and **(3) reliability-aware aggregation** scheme to prioritize high-quality client updates.

The training process that integrates these components begins with a brief **server warm-up phase**, where the server's small labeled ID set is used to initialize the model and server-only BatchNorm statistics. Following this, the main federated training proceeds in rounds. A single round $t$ consists of a carefully coordinated sequence:

(1) **Server Broadcast:** The server distributes the current global model $\theta^t$ and a set of global pivots $\{\mu_k^{t-1}\}$ (one for each ID class), generated from the stable EMA model of the previous round.

(2) **Client Training:** Sampled clients perform local training on their unlabeled data, guided by the received model and pivots (Sec. 4.3).

(3) **Server Update:** The server aggregates client models using our Reliability-Aware Aggregation (RAA) scheme (Sec. 4.4), fine-tunes the result, and updates its EMA model to generate new pivots $\{\mu_k^t\}$ for the next round (Sec. 4.2).

This iterative process ensures that local updates are stabilized by server-side supervision, temporal smoothing, and pivot-guided representation learning. The entire procedure is formally detailed in Pseudo Code in the Appendix C.

## 4.1 OVERALL FRAMEWORK AND LEARNING OBJECTIVES

We adopt a learning framework where the server and clients have disjoint roles. The server trains on its small, clean labeled dataset, while clients train exclusively on their large, unlabeled open-set data streams. The overall learning objective combines a standard closed-set classifier and an OOD detector, *with loss terms separated by data source*.

- **Server-Side (Labeled Data):** On the server, we use the supervised cross-entropy loss ($\mathcal{L}_{ce}$) for the ID classifier and a one-vs-all (OVA) loss ($\mathcal{L}_{od}^{OVA}$) for the OOD detector, following SSB (Fan et al., 2023).

$$\mathcal{L}_{server} = \mathcal{L}_{ce} + \lambda_{od}\, \mathcal{L}_{od}^{OVA}$$

- **Client-Side (Unlabeled Data):** On clients, the learning objective is threefold. First, following FixMatch (Sohn et al., 2020), we apply a consistency regularization loss ($\mathcal{L}_{con}$) for the ID classifier. Second, we employ unlabeled data objectives ($\mathcal{L}_{OOD}^{unlab}$) for the OOD detector, as proposed in OpenMatch and SSB (Saito et al., 2021; Fan et al., 2023). Finally, and most crucially, we introduce our **pivot-guided alignment loss ($\mathcal{L}_{align}$)**. The final client-side loss is a weighted sum of these three components:

$$\mathcal{L}_{client} = \lambda_{con}\mathcal{L}_{con} + \lambda_{align}\mathcal{L}_{align} + \lambda_{od}\mathcal{L}_{OOD}^{unlab}$$

where $\mathcal{L}_{OOD}^{unlab}$ combines entropy minimization, soft open-set consistency regularization (SOCR), and a pseudo-negative loss:

$$\mathcal{L}_{OOD}^{unlab} = \lambda_{em}\mathcal{L}_{od}^{em} + \lambda_{SOCR}\mathcal{L}_{od}^{SOCR} + \lambda_{neg}\mathcal{L}_{od}^{neg}$$

## 4.2 ROUND-WISE EMA (R-EMA)

To damp round-to-round drift observed in our empirical study, the server maintains a **round-wise EMA (R-EMA)** model $\theta_{EMA}$. After aggregating client updates and server fine-tuning at round $t$ ($\theta_{agg}^t \rightarrow \theta_{fine}^t$), we update

$$\theta_{EMA}^t = \alpha\, \theta_{EMA}^{t-1} + (1 - \alpha)\, \theta_{fine}^t, \quad \alpha \in [0, 1).$$

Here, $\alpha$ controls the EMA memory horizon (larger $\alpha \Rightarrow$ slower adaptation). Unlike iteration-level EMA in centralized SSL, R-EMA updates once per *communication round*, which reduces oscillation and stabilizes the feature geometry. Because each round summarizes many local steps, typical iteration-level decays (e.g., 0.99–0.999) are overly inertial; we therefore adopt a *moderate* decay and use $\alpha = 0.9$ by default (see Sec. 5.3).

R-EMA serves two roles: (i) it is our **final inference model**—temporal smoothing improves generalization; and (ii) it provides a **stable backbone for class pivots** used in client-side alignment. We deliberately *do not* use the EMA model to emit pseudo-labels or transmit it to clients: temporal averaging can damp softmax confidences (undesirable for high-threshold pseudo-labeling), and sending an extra model substantially increases downlink bandwidth. Hence, EMA is used exclusively on the server for pivot generation and final evaluation.

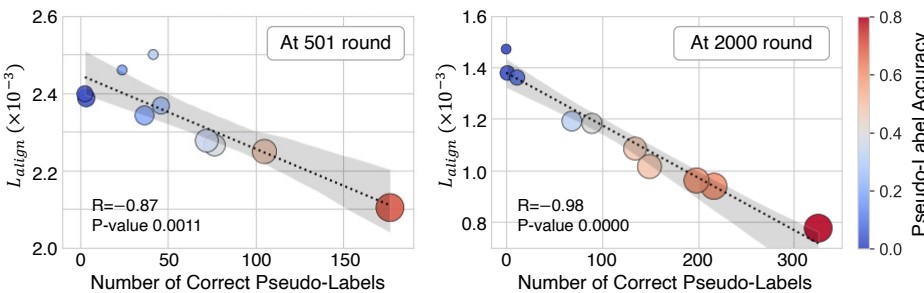

Figure 3: **Alignment loss as a strong indicator of client reliability.** Each point represents a client at an early (round 501) and late (round 2000) stage of training. The plots show a clear negative correlation between the average alignment loss (y-axis) and pseudo-label quality, measured by the number of correct pseudo-labels (x-axis) and accuracy (color). The marker size further reflects the total number of accepted pseudo-labels. This empirically validates using the inverse of $L_{\text{align},c}$ as a client weight in our RAA scheme.

### 4.3 PIVOT-GUIDED OPEN-SET ALIGNMENT

Direct training on client-side pseudo-labels suffers from **pseudo-label brittleness**: under Non-IID open-set splits, small errors are amplified across rounds, leading to severe mislabeling and rapid error propagation. We therefore guide local representation learning with *server-anchored* signals so that only trustworthy evidence shapes the global model. Following prototype-based alignment (Choi et al., 2025), we adapt it to the federated labels-at-server setting, where clients lack labeled anchors, confidence is unstable, and heterogeneous OOD distributions further complicate selection.

**Global pivots.** From the server's small labeled set and its round-wise EMA (R-EMA) model, we compute **global pivots** $\{\mu_k\}$ once per round and distribute them to clients as stable class references.

**Dual-gate selection.** A client sample is aligned only if both the closed-set classifier and the OOD detector agree it is a high-confidence ID. For an embedding $z_i$ with predicted label $\hat{k} = \arg\max_k p_{i,k}$, we use

$$\Phi_i = \mathbb{1}\big(p_{i,\hat{k}} > \tau_{ID} \ \wedge \ \varphi_{i,\hat{k}}^{ID} > \eta_{ID}\big).$$

Here, $p_{i,k}$ is the classifier's softmax posterior for class $k$, and $\varphi_{i,k}^{ID}$ is the OOD detector's inlier score (OVA positive-class probability). The thresholds $\tau_{ID}, \eta_{ID} \in (0,1)$ control gate strictness; in practice we set $\eta_{ID} = 0.9$ for reliability and use a high-confidence *softmax* threshold $\tau_{ID} \in [0.95, 0.99]$, a range commonly adopted for confidence-based pseudo-labeling (e.g., FixMatch). The gate is computed once by the server each round and held fixed during a client's local steps to avoid drift.

**Alignment loss and its effect.** Passing samples ($\Phi_i{=}1$) are *attracted* toward their pivots, while non-passing ones ($\Phi_i{=}0$) receive a mild *angular repulsion* from all pivots via the normalization term:

$$\mathcal{L}_{\text{align}}(z_i) = -\Phi_i \frac{\text{sim}(z_i, \mu_{\hat{k}})}{T} + \log \sum_{j=1}^{K} \exp\left(\frac{\text{sim}(z_i, \mu_j)}{T}\right). \tag{1}$$

Here, sim is cosine similarity and $T$ is a temperature. Thus, confident ID embeddings are selectively pulled toward global references, whereas uncertain/OOD embeddings are *discouraged from premature commitment*. Moreover, with unit-normalized features, maximizing cosine to $\mu_k$ is equivalent to minimizing $\|\nu_c - \mu_k\|^2$ for client class means $\nu_c$; since $\mathcal{D}_k = \frac{1}{C}\sum_c \|\nu_c - \bar{\nu}_k\|^2 \leq \frac{1}{C}\sum_c \|\nu_c - \mu_k\|^2$, optimizing Eq. 1 reduces an upper bound on inter-client variance.

### 4.4 RELIABILITY-AWARE AGGREGATION (RAA)

Standard federated averaging weights clients by their data quantity ($n_c$), but in semi-supervised settings the *quality* of updates is more critical. We therefore introduce **Reliability-Aware Aggregation (RAA)**, a quality-over-quantity scheme.

RAA uses the client's average alignment loss $\mathcal{L}_{\text{align},c}$ as an *inverse proxy for reliability*, grounded in the structure of Eq. 1. For a correctly gated ID sample, the attractive term largely cancels the

normalization term, yielding a small loss; by contrast, (i) a misclassified ID sample retains a large normalization term (dominated by its true pivot), and (ii) a mis-gated OOD sample has no matching pivot, so the cancellation fails—both cases produce much larger losses. Thus $\mathcal{L}_{\text{align},c}$ tracks pseudo-label quality.

Formally, for client $c$ with pseudo-label accuracy $q_c$,

$$\mathbb{E}[\mathcal{L}_{\text{align},c}] = q_c\,\mathbb{E}[\mathcal{L}_{\text{align},c}\,|\,\text{correct}] + (1 - q_c)\,\mathbb{E}[\mathcal{L}_{\text{align},c}\,|\,\text{wrong}] \approx (1 - q_c)\,\Delta,\ \Delta > 0,$$

so lower $\mathcal{L}_{\text{align},c}$ indicates higher reliability. We then set the aggregation weights $\tilde{n}_c$ and update $\theta_{\text{agg}}^{t+1}$:

$$\tilde{n}_c = \frac{\kappa}{\mathcal{L}_{\text{align},c} + \epsilon}, \qquad \theta_{\text{agg}}^{t+1} = \frac{\sum_{c \in S_t} \tilde{n}_c\,\theta_c^t}{\sum_{c \in S_t} \tilde{n}_c}, \tag{2}$$

where $S_t$ is the set of participating clients at round $t$, and $\kappa$ (scale) and $\epsilon$ (stability) are constants. RAA up-weights clients with cleaner pseudo-labels and down-weights noisy ones, curbing confirmation bias over rounds. The inverse relationship between $\mathcal{L}_{\text{align},c}$ and pseudo-label quality is borne out empirically in Figure 3.

## 5 EXPERIMENTS

### 5.1 EXPERIMENTAL SETUP

**Evaluation Protocol and Baselines.** We evaluate models on inlier classification accuracy (Acc.) and OOD detection AUROC. To assess generalization, we report results on both **seen** OODs (present during client training) and a diverse set of **unseen** OOD datasets, and we summarize with an *overall AUROC* defined as the unweighted mean of the seen AUROC and the average AUROC over all unseen OOD sets. Baselines include FedSSL methods (SemiFL and FedFixMatch) and federated versions of centralized OSSL methods (FedSCOMatch, FedProSub, and FedSSB).

**Datasets and backbones.** We adopt open-set semi-supervised splits with few labels per ID class. On **FashionMNIST** (Xiao et al., 2017), we use 6 clothing classes as ID and 4 accessories as OOD, with 40 labeled samples per ID class; the backbone is a lightweight CNN with three convolutional blocks. On **CIFAR-10** (Krizhevsky et al., 2009), we use 6 animals as ID and 4 vehicles as OOD, evaluating with 20 and 40 labeled samples per ID class. On **CIFAR-100**, we consider two ID/OOD ratios (55/45 and 80/20), using 40 labeled samples per ID class in both settings. For CIFAR-10/100, we employ WRN-28-2 (Zagoruyko, 2016) for both the classifier and OOD detector heads. For unseen OOD evaluation, we use cross-dataset pairs (CIFAR-10→CIFAR-100, CIFAR-100→CIFAR-10) and additionally SVHN (Netzer et al., 2011), LSUN (Yu et al., 2015), ImageNet (Deng et al., 2009) (resized), and Gaussian noise.

**Federated Setting and Implementation.** We simulate a cross-device FL scenario with 100 clients and a 10% participation ratio per round, under IID and Non-IID partitions (Dirichlet label skew with $\alpha \in \{0.3,\ 0.1\}$). Training consists of a server warm-up followed by federated rounds: FashionMNIST uses a 300-epoch warm-up and 1100 rounds; CIFAR-10/100 use a 500-epoch warm-up and 2500 rounds. Experiments are implemented on Flower (Beutel et al., 2020); full hyperparameters and implementation details are provided in the Appendix A.2.

### 5.2 MAIN RESULTS

As shown in Table 1, *OpenFL* delivers state-of-the-art performance on the challenging CIFAR-10/100 benchmarks across IID and Non-IID partitions, improving *both* closed-set accuracy and overall AUROC. FedSSL baselines (SemiFL, FedFixMatch) achieve reasonable accuracy but weaker OOD detection, whereas federated versions of centralized OSSL methods (FedSCOMatch, FedProSub) tend to be unstable under Non-IID splits. FedSSB remains the strongest AUROC competitor yet consistently trades off inlier accuracy. Training dynamics further show smooth convergence with reduced oscillation (Appendix B.1, Fig. 5), consistent with the stabilizing effects of R-EMA and pivot-guided alignment.

On FashionMNIST, SemiFL and FedFixMatch slightly leads overall. This dataset offers a relatively easy ID/seen-OOD separation (clothing vs. accessories), so pseudo-labels are already reliable and classic SSL objectives (e.g., FixMatch-style consistency) excel. Even so, *OpenFL* stays competitive and remains robust under stronger heterogeneity (e.g., $\text{Dir}(0.1)$), while on CIFAR-10/100 the advantage widens as Non-IID skew increases.

Table 1: **Performance comparison of *OpenFL* against baselines on all datasets and client partitioning schemes.** We report closed-set classification accuracy (Acc., %) and overall OOD detection performance (overall AUC). All results are the mean $\pm$ standard deviation over 3 runs with different random seeds. The **best** results are in bold, and the second-best are underlined.

| Client Partitioning | SemiFL | | FedFix | | FedSCO | | FedProSub | | FedSSB | | *OpenFL* | |
|---|---|---|---|---|---|---|---|---|---|---|---|---|
| | Acc. | AUROC | Acc. | AUROC | Acc. | AUROC | Acc. | AUROC | Acc. | AUROC | Acc. | AUROC |
| **CIFAR-10 6/4, lb120 (20 Labels/Class)** | | | | | | | | | | | | |
| IID | 38.9 ±9.7 | 45.3 ±7.4 | 77.1 ±2.1 | 45.5 ±5.9 | 65.2 ±5.2 | 44.4 ±4.9 | 24.6 ±12.3 | 39.5 ±3.2 | 81.4 ±3.3 | 91.5 ±3.4 | **83.9** ±2.2 | **93.1** ±1.7 |
| $Dir(0.3)$ | 27.4 ±18.5 | 50.3 ±7.7 | 61.8 ±2.2 | 49.1 ±2.6 | 54.0 ±4.7 | 45.8 ±6.2 | 17.2 ±0.6 | 40.6 ±5.7 | 64.4 ±1.1 | **79.2** ±8.6 | **67.3** ±3.6 | 78.9 ±5.6 |
| $Dir(0.1)$ | 16.7 ±0.0 | 49.9 ±3.7 | 50.9 ±2.0 | 48.7 ±6.8 | 45.2 ±2.6 | 51.2 ±2.3 | 22.9 ±5.6 | 47.2 ±8.6 | 51.6 ±2.8 | 71.2 ±1.3 | **55.1** ±3.5 | **74.5** ±3.0 |
| **CIFAR-10 6/4, lb240 (40 Labels/Class)** | | | | | | | | | | | | |
| IID | 50.3 ±31.0 | 38.1 ±14.2 | 87.5 ±0.2 | 66.7 ±7.5 | 73.1 ±1.3 | 45.9 ±3.4 | 38.0 ±3.1 | 27.2 ±1.9 | 87.7 ±1.1 | 91.7 ±1.4 | **88.3** ±0.3 | **92.7** ±0.5 |
| $Dir(0.3)$ | 51.9 ±30.7 | 51.5 ±9.6 | 73.3 ±1.8 | 54.1 ±4.3 | 61.6 ±1.1 | 48.3 ±3.1 | 32.8 ±7.6 | 34.1 ±5.2 | 74.8 ±3.0 | 87.8 ±3.4 | **76.8** ±3.3 | **88.3** ±3.8 |
| $Dir(0.1)$ | 16.7 ±0.0 | 44.1 ±11.8 | 62.8 ±0.7 | 68.8 ±4.4 | 54.3 ±2.4 | 51.7 ±3.6 | 31.2 ±13.5 | 41.1 ±3.7 | 65.5 ±2.2 | **74.9** ±1.8 | **67.5** ±1.3 | 73.1 ±5.7 |
| **CIFAR-100 55/45, lb2200 (40 Labels/Class)** | | | | | | | | | | | | |
| IID | 63.1 ±0.4 | 68.3 ±0.6 | 66.4 ±0.8 | 68.0 ±1.1 | 62.5 ±0.4 | 72.1 ±3.0 | 25.2 ±2.7 | 49.7 ±3.5 | 68.1 ±0.6 | **75.5** ±2.3 | **69.0** ±0.7 | 74.8 ±3.6 |
| $Dir(0.3)$ | 61.2 ±0.5 | 67.6 ±0.3 | 63.6 ±1.0 | 65.3 ±2.8 | 58.6 ±0.4 | 70.6 ±0.3 | 27.6 ±1.9 | 48.3 ±1.8 | 65.4 ±0.7 | **75.0** ±2.0 | **66.1** ±0.6 | **75.0** ±2.3 |
| $Dir(0.1)$ | 59.5 ±1.1 | 65.2 ±1.7 | 60.3 ±1.2 | 65.7 ±0.7 | 55.6 ±0.1 | 70.4 ±0.2 | 28.5 ±2.9 | 50.0 ±0.8 | 61.8 ±1.2 | 71.5 ±2.6 | **62.8** ±0.5 | **74.5** ±3.0 |
| **CIFAR-100 80/20, lb3200 (40 Labels/Class)** | | | | | | | | | | | | |
| IID | 56.8 ±1.0 | 66.1 ±1.2 | 59.7 ±0.3 | 62.2 ±0.6 | 57.3 ±0.3 | 66.3 ±0.3 | 23.0 ±0.4 | 49.7 ±0.3 | 59.6 ±0.6 | **75.6** ±0.9 | **61.1** ±0.8 | 74.6 ±2.4 |
| $Dir(0.3)$ | 54.1 ±0.5 | 64.7 ±0.2 | 56.3 ±0.5 | 60.5 ±0.9 | 53.2 ±0.5 | 65.2 ±1.4 | 21.5 ±4.4 | 50.9 ±1.4 | 55.0 ±3.3 | 71.4 ±3.9 | **58.0** ±0.7 | **75.4** ±1.2 |
| $Dir(0.1)$ | 52.4 ±0.6 | 61.2 ±1.8 | 52.8 ±0.6 | 60.7 ±2.8 | 49.3 ±0.2 | 66.8 ±1.4 | 23.3 ±1.2 | 50.3 ±0.2 | 53.4 ±0.4 | 71.8 ±0.9 | **55.0** ±0.5 | **72.7** ±3.4 |
| **FashionMNIST, lb240 (40 Labels/Class)** | | | | | | | | | | | | |
| IID | **81.5** ±0.0 | 59.2 ±25.8 | 80.7 ±2.5 | 86.0 ±2.6 | 76.7 ±2.1 | 47.7 ±9.6 | 75.7 ±0.7 | 14.5 ±4.6 | 78.5 ±2.6 | **93.6** ±2.2 | 78.3 ±6.7 | 86.6 ±10.8 |
| $Dir(0.3)$ | 76.1 ±0.0 | 65.5 ±5.9 | 77.9 ±1.4 | **86.9** ±3.2 | 74.5 ±3.3 | 46.5 ±5.8 | 74.1 ±1.3 | 17.8 ±2.0 | 76.2 ±2.2 | 81.6 ±21.5 | **79.2** ±1.0 | 86.1 ±13.6 |
| $Dir(0.1)$ | 73.1 ±0.0 | 55.5 ±3.8 | **75.5** ±1.6 | 83.9 ±3.1 | 72.3 ±3.2 | 48.9 ±10.8 | 72.2 ±1.9 | 12.7 ±2.1 | 75.1 ±3.6 | 87.9 ±0.2 | 75.3 ±3.1 | **95.3** ±1.3 |

Table 2: **Effect of R-EMA and pivot source.** Results on CIFAR-10 (6/4, lb240) and CIFAR-100 (80/20, lb3200) comparing: w/o EMA ($\alpha = 0$), EMA with $\alpha \in \{0.95, 0.9, 0.85\}$, and a variant that keeps EMA ($\alpha = 0.9$) but computes pivots from the fine-tuned (non-EMA) server model used for the current training.

| Dataset | w/o EMA $\alpha = 0$ | | EMA $\alpha = 0.95$ | | EMA $\alpha = 0.9$ | | EMA $\alpha = 0.85$ | | non-EMA Pivots | |
|---|---|---|---|---|---|---|---|---|---|---|
| | Acc. | AUROC | Acc. | AUROC | Acc. | AUROC | Acc. | AUROC | Acc. | AUROC |
| **CIFAR-10 6/4, lb240** | 75.9 | 90.0 | 76.6 | **93.4** | **77.6** | 92.3 | 74.6 | 91.4 | 76.3 | 93.0 |
| **CIFAR-100 80/20, lb3200** | 58.1 | 71.2 | 59.2 | 72.3 | **59.7** | **76.4** | 58.2 | 73.7 | 59.3 | 72.3 |

## 5.3 ABLATION

**Ablation on Round-EMA and Pivot Source.** We ablate two factors—temporal smoothing via EMA and the source of class pivots. Enabling R-EMA consistently improves AUROC and typically boosts closed-set accuracy, indicating that per-communication-round smoothing stabilizes representations under federated noise. Among decay values, a *moderate* setting ($\alpha = 0.9$) offers the best balance, whereas a higher decay (0.95) is overly inertial and a lower one (0.85) under-smooths. Keeping EMA at $\alpha = 0.9$ but deriving pivots from the non-EMA (fine-tuned) model weakens performance on the harder CIFAR-100 split and introduces a mild Acc–AUROC trade-off on CIFAR-10, suggesting that EMA-derived features provide better-calibrated anchors. We therefore adopt $\alpha = 0.9$ with EMA-derived pivots by default.

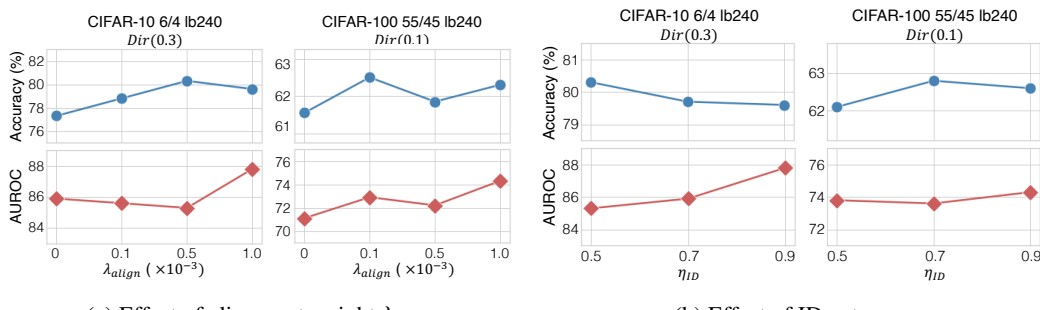

(a) Effect of alignment weight $\lambda_{\text{align}}$.  (b) Effect of ID gate $\eta_{\text{ID}}$.

Figure 4: **Sensitivity of Pivot-Guided Alignment.** Accuracy (blue) and AUROC (red) versus (a) $\lambda_{\text{align}}$ and (b) $\eta_{\text{ID}}$ under Dirichlet splits (CIFAR-10 6/4, Dir 0.3; CIFAR-100 55/45, Dir 0.1).

Table 3: **Effect of reliability-aware aggregation.** Comparison with FedAvg on CIFAR-10 (6/4; lb120/lb240) and CIFAR-100 (55/45; lb2200, 80/20; lb3200); FashionMNIST (6/4; lb240) included.

| | CIFAR-10 6/4, lb120 | | CIFAR-10 6/4, lb240 | | CIFAR-100 55/45, lb2200 | | CIFAR-100 80/20, lb3200 | | FashionMNIST 6/4, lb240 | |
|---|---|---|---|---|---|---|---|---|---|---|
| | Acc. | AUROC | Acc. | AUROC | Acc. | AUROC | Acc. | AUROC | Acc. | AUROC |
| FedAvg | 65.6 | 83.2 | 74.9 | 91.1 | 62.2 | 71.6 | 59.1 | 74.8 | 78.2 | 92.3 |
| **RAA** | 71.5 | 83.9 | 77.7 | 92.3 | 62.9 | 71.2 | 59.7 | 76.4 | 79.1 | 95.1 |
| | ↑5.9 | ↑0.7 | ↑2.8 | ↑1.2 | ↑0.7 | ↓0.4 | ↑0.6 | ↑1.6 | ↑0.9 | ↑2.8 |

A natural question is whether simply adding EMA to a strong baseline suffices. Applying the same *R-EMA* to FedSSB improves stability and AUROC (Appendix Fig. 6) and yields higher numbers than plain FedSSB (Appendix B.2 Table 4); however, *OpenFL* still outperforms *SSB with R-EMA* on both accuracy and overall AUROC across the evaluated splits, including the more challenging CIFAR-100 (80/20). This indicates that our gains do not stem from EMA alone—the combination of EMA-stabilized pivots, pivot-guided open-set alignment, and RAA is needed to attain the final performance.

**Sensitivity of Pivot-Guided Alignment ($\lambda_{\text{align}}$, $\eta_{\text{ID}}$).** Figure 4 studies the two knobs of our pivot-guided alignment. Any $\lambda_{\text{align}} > 0$ improves Acc. and AUROC over $\lambda_{\text{align}} = 0$. On CIFAR-10, accuracy peaks near $\lambda_{\text{align}} \approx 0.5$ while AUROC keeps rising to 1.0; CIFAR-100 prefers a smaller weight (about 0.1). For the ID gate $\eta_{\text{ID}}$, stricter gating increases AUROC on both datasets with only a mild accuracy trade-off on CIFAR-10. Both knobs exhibit a broad stability plateau, with modest variation for $\lambda_{\text{align}} \in [0.1, 1.0]$ and $\eta_{\text{ID}} \in [0.7, 0.9]$. Although the OOD detector's natural decision boundary is 0.5, we set a high ID gate ($\eta_{\text{ID}} \approx 0.9$) to ensure reliable alignment; this mainly changes the quantity, not the quality, of aligned samples. We use $\lambda_{\text{align}} = 0.5$ and $\eta_{\text{ID}} = 0.9$ by default.

**Effect of Reliability-Aware Aggregation.** We ablate reliability-aware aggregation (RA-Agg), which reweights clients by the inverse of their average alignment loss $L_{\text{align},c}$, against FedAvg. On CIFAR-10, RA-Agg consistently improves both accuracy and AUROC, indicating that emphasizing clients with cleaner pseudo-labels benefits the global model. On CIFAR-100, gains are smaller or mixed: with many classes, dual-gated positives are sparser and the log-sum-exp term dominates, compressing the across-client spread of $L_{\text{align}}$ so the reweighting approaches near-uniform averaging; class coverage also becomes more critical. Overall, RA-Agg is most impactful when inter-client pseudo-label quality varies widely, while remaining a safe, sometimes positive modification in the 100-class regime.

## 6 CONCLUSION

We presented, to our knowledge, the first systematic study of FOSSL in the *labels-at-server* setting, revealing *pseudo-label brittleness* on clients and amplified instability under OOD heterogeneity. Building on these findings, we proposed *OpenFL*, combining Round-wise EMA, Pivot-guided Open-set Alignment, and Reliability-Aware Aggregation. Across datasets and partitions (IID and Non-IID), *OpenFL* achieves strong inlier accuracy and overall AUROC with stable convergence. **Limitations.** Our method still relies on globally fixed hyperparameters (e.g., $\tau_{ID}$, $\eta_{ID}$, $\alpha$, $\lambda_{\text{align}}$); future work includes adaptive client-/round-aware tuning and stronger robustness to adversarial or non-stationary clients.

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

# APPENDIX

## A  IMPLEMENTATION DETAILS

### A.1  BASELINES

For a fair comparison, we implemented all baselines in the federated learning setting by incorporating commonly used techniques, such as alternative training and pseudo-labeling from SemiFL Diao et al. (2022) and static batch normalization (StaticBN) from HeteroFL Diao et al. (2021). SemiFL is implemented directly based on the original code from the authors, while the other baselines are re-implemented on top of the Flower framework Beutel et al. (2020) to ensure consistency with ours. The implementation details of each baseline are described as follows.

**SemiFL.** *Server:* trains on its small labeled dataset with supervised cross-entropy loss. *Clients:* receive the global model and optimize consistency loss using pseudo-labels from the server model, following the original SemiFL design.

**FedFixMatch.** *Server:* minimizes supervised cross-entropy loss on the labeled ID set. *Clients:* apply FixMatch logic with weak/strong augmentations. For each unlabeled sample, if the weakly-augmented prediction exceeds a confidence threshold $\tau$, the client enforces a consistency loss on the strongly-augmented view: $\mathcal{L}_{\text{cons}} = \mathbf{1}(\max p(y|x^w) > \tau) \cdot \text{CE}(p(y|x^s), \hat{y})$. Thus, FedFixMatch pairs server-side supervision with client-side consistency training on pseudo-labeled data.

**FedSCOMatch.** *Server:* trains with cross-entropy loss on labeled ID data. *Clients:* Clients adapt a dual-stream strategy: the open-set stream applies pseudo-labeling over $K + 1$ classes, while the close-set stream applies pseudo-labeling over only $K$ ID classes. An OOD memory queue stores low-MSP samples for $(K + 1)th$ class.

**FedProSub.** *Server:* The Server maintains ID class prototypes and minimized supervised cross-entropy loss on labeled data. Subspace scores are computed via QR decomposition, and only the Beta distribution parameters for the ID distribution $(\alpha_1, \beta_1)$ are updated. Although the prototypes are described as EMA-updated. *Clients:* Each client computes subspace scores through QR decomposition and applies FixMatch-style pseudo-labeling weighted by $p(ID|subspace score)$. Clients optimize tree losses: $L_{xep}$(cross-entropy pseudo-label loss), $L_{sub}$(subspace separation loss), and $L_{us}$(unsupervised similarity loss). Unlike the server, clients update the Beta distribution parameters for both ID $(\alpha_1, \beta_1)$ and OOD $(\alpha_2, \beta_2)$

**FedSSB.** *Server:* jointly trains the classifier with cross-entropy loss and an auxiliary one-vs-all (OVA) classifier with OVA loss, enabling OOD-aware supervision. *Clients:* generate pseudo-labels with the server model. The classifier head is trained with consistency loss, while the OVA head is optimized with soft consistency loss, entropy minimization, and negative mining (following SkipAlign). This combination improves OOD discrimination under open-set federated conditions.

### A.2  HYPER PARAMETER DETAILS

Across all CIFAR settings, the server and clients adopt SGD as the optimizer with momentum fixed at 0.9 and weight decay at 0.0005. Training is conducted for a total of 2500 communication rounds. On the server side, every setting involves 500 warmup epochs followed by 3 fine-tuning epochs, while clients perform 5 local training epochs. The client participation ratio is consistently set to 0.1, with a total of 100 clients available. A cosine-style scheduler is enabled in all cases, though the minimum learning rate varies depending on the dataset.

**CIFAR-10, 6/4, lb120 and CIFAR-10, 6/4, lb240** Both the server and clients use a learning rate of 0.01. The scheduler lowers the rate to a minimum of 0.001. The server employs a batch size of 30 for training and 200 for testing, while the client batch size is 32.

**CIFAR-100, 55/45, lb2200 and CIFAR-100, 80/20, lb3200** The learning rate is increased to 0.03 for both server and clients, with the cosine scheduler reducing it to a minimum of 0.005. The server's batch sizes are larger, namely 100 for training, while the client uses a batch size of 64.

For both CIFAR-10 and CIFAR-100 datasets, he weak augmentation includes resizing, random cropping with reflection padding, random horizontal flipping, and normalization. The strong augmentation follows the same procedure but additionally applies RandAugmentMC with two operations and magnitude 10. For evaluation, we only use resizing and normalization.

**fashionmnist, 6/4, lb240** Both server and clients use SGD with a learning rate of 0.01 and a cosine scheduler with minimum 0.001. Training runs for 1100 rounds, the server runs 300 warmup epochs and 3 fine-tuning epochs, while clients train for 5 local epochs. The server batch size is 30 for training, and the client batch size is 32. The client participation ratio is consistently set to 0.1, with a total of 100 clients available.

For the implementation of other baselines, we aligned general settings such as batch size and the number of rounds with those of *OpenFL* whenever applicable, while applying minor adjustments when baseline-specific requirements were necessary.

## B    EXTENDED EXPERIMENTAL RESULTS

### B.1    TRAINING DYNAMICS AND STABILITY

Figure 5 plots per-round closed-set accuracy and overall AUROC on CIFAR-100 (80/20) under IID and Dirichlet partitions. Across all splits, *OpenFL* converges smoothly with low oscillation; Non-IID primarily slows accuracy growth, while AUROC remains robust and keeps improving, finishing on par with—or slightly above—the IID case. These dynamics are consistent with the stabilizing effects of R-EMA and pivot-guided alignment described in Sec. 4.

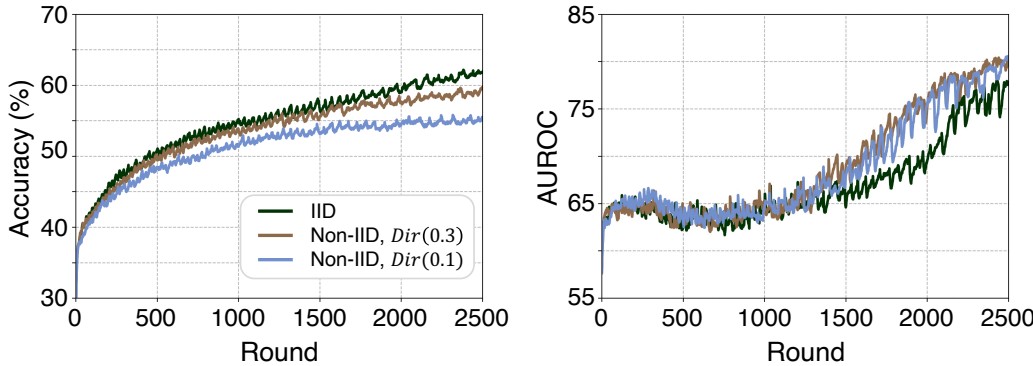

Figure 5: **Training dynamics on CIFAR-100 (80/20).** Accuracy (left) and overall AUROC (right) vs. round under IID and Dirichlet partitions. R-EMA and pivot-guided alignment yield stable convergence; heterogeneity mainly slows Acc., while AUROC remains robust and continues to improve.

### B.2    APPLYING ROUND-WISE EMA TO FEDSSB

To test whether gains come merely from temporal smoothing, we apply the same round-wise EMA (R-EMA, $\alpha=0.9$) to FedSSB. As shown in Fig. 6, EMA reduces oscillation and modestly improves both Acc. and overall AUROC. Quantitatively (Table 4), FedSSB with R-EMA outperforms vanilla FedSSB, yet *OpenFL* still achieves higher accuracy and AUROC across the evaluated splits, indicating that EMA alone is insufficient without our pivot-guided alignment and reliability-aware aggregation.

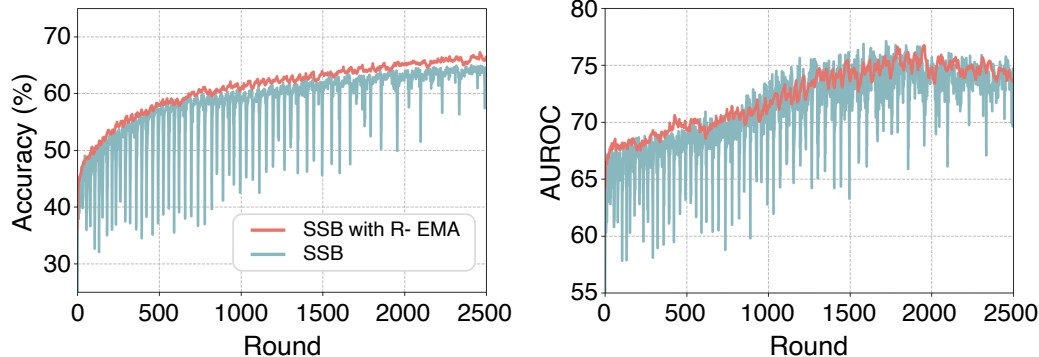

Figure 6: **FedSSB with vs. without Round-wise EMA (R-EMA, $\alpha$=0.9).** CIFAR-100 55/45 ($Dir(0.3)$) training dynamics: accuracy (left) and overall AUROC (right) vs. round. R-EMA reduces round-to-round oscillation and yields smoother, higher curves. All other settings are identical; only EMA is toggled.

Table 4: **FedSSB, FedSSB with R-EMA, and *OpenFL*.** Accuracy (Acc.) and overall AUROC on CIFAR-10 (6/4, lb240) and CIFAR-100 (80/20, lb3200) under IID and Dirichlet partitions. Round-wise EMA consistently improves FedSSB, but *OpenFL* remains best in both Acc. and AUROC. Results are mean $\pm$ std over three seeds.

| | IID | | Dir(0.3) | | Dir(0.1) | |
|---|---|---|---|---|---|---|
| | Acc. | AUROC | Acc. | AUROC | Acc. | AUROC |
| **CIFAR-10 6/4 lb240** | | | | | | |
| SSB | 87.7 | 91.7 | 74.8 | 87.8 | 65.5 | 74.9 |
| | ±1.1 | ±1.4 | ±3.0 | ±3.4 | ±2.2 | ±1.8 |
| SSB EMA | 87.9 | 92.1 | 75.5 | 87.8 | 66.4 | 73.6 |
| | ±0.5 | ±1.3 | ±0.8 | ±0.6 | ±1.4 | ±1.8 |
| *OpenFL* | 88.3 | 92.7 | 76.8 | 88.3 | 67.5 | 73.1 |
| | ±0.3 | ±0.5 | ±3.3 | ±3.8 | ±1.3 | ±5.7 |
| **CIFAR-100 80/20 lb3200** | | | | | | |
| SSB | 59.6 | 75.6 | 55.0 | 71.4 | 53.4 | 71.8 |
| | ±0.6 | ±0.9 | ±3.3 | ±3.9 | ±0.4 | ±0.9 |
| SSB EMA | 60.1 | 75.3 | 57.3 | 72.6 | 54.5 | 70.6 |
| | ±0.6 | ±1.9 | ±0.2 | ±1.7 | ±0.9 | ±1.2 |
| *OpenFL* | 61.1 | 74.6 | 58.0 | 75.4 | 55.0 | 72.7 |
| | ±0.8 | ±2.4 | ±0.7 | ±1.2 | ±0.5 | ±3.4 |

## C ALGORITHM PSEUDO CODE

---

**Algorithm 1:** OpenFL (Main Process and Client Procedure)

---

**Input:** Labeled server data $\mathcal{D}_l$; Local unlabeled data of client c $\mathcal{D}_{u,c}$; Total communication
rounds $T$; Client sampling fraction $\xi$; Total clients $C$; Initial model parameters
$\theta_g, \theta_{EMA}$; Warmup epoch $E_w$; EMA momentum $\alpha$; Round to start applying alignment
loss $T_{align}$

**Output:** Final trained global model parameters $\theta_g^T$

**System executes:**

    // Phase 1:  Initial Server Warmup
    $\theta_g^0, \theta_{EMA}^0, \mu^0 \leftarrow$ **ServerWarmup**$(\theta_g, \theta_{EMA})$
    // Phase 2:  Federated Learning Rounds
    **for** *round $t = 0, 1, \ldots, T-1$* **do**
        // Step 2.1:  Client Selection & Training
        $S_t \leftarrow$ (Randomly sample $\xi \cdot C$ clients from $C$ clients)
        **for** *each client $c \in S_t$ in parallel* **do**
            Distribute global model $\theta_g^t$ and global pivots $\mu^t$ to client $c$
            Request local training: $(\theta_c^{t+1}, \mathcal{L}_{align,c}) \leftarrow$ **ClientUpdate**$(c, \theta_g^t, \mu^t, t)$

        // Step 2.2:  Reliability-Aware Aggregation
        Receive updated models $\{\theta_c^{t+1}\}$ and align loss $\{\mathcal{L}_{align,c}\}$ from clients in $S_t$
        $\theta_g^{t+1} \leftarrow$ **ServerAggregation**$(\{(\theta_c^{t+1}, \mathcal{L}_{align,c})\})$
        $\theta_g^{t+1}, \theta_{EMA}^{t+1}, \mu^{t+1} \leftarrow$ **ServerFineTune**$(\theta_g^{t+1}, \theta_{EMA}^t, \mu^t, t+1)$

**Procedure** *ClientUpdate*$(c, \theta_g^t, \mu^t, t)$

    // Stage 1:  Pre-computation of Masks(inference mode)
    **for** $x \in \mathcal{D}_{u,c}$ **do**
        $h \leftarrow \mathcal{F}(x)$                         // Extract features
        $p \leftarrow \text{Softmax}(Classifier(h))$       // ID classifier head
        $\varphi \leftarrow OD(h)$                // OOD detector head
        Compute masks: $\Phi_x^{dual} \leftarrow (p \geq \tau_{ID}$ and $\varphi_{ID} \geq \eta_{ID}), \Phi_x^{neg} \leftarrow (\varphi_{OOD} \geq \tau_{neg})$

    // Stage 2:  Local Training
    **for** *epoch $= 1$ to $E_c$* **do**
        Sample a batch $\mathcal{B}$ from $\mathcal{D}_{u,c}$
        Let $\mathcal{B}_{dual} \leftarrow \{x \in \mathcal{B} \mid \Phi_x^{dual} = 1\}$
        Let $\mathcal{B}_{neg} \leftarrow \{x \in \mathcal{B} \mid \Phi_x^{neg} = 1\}$
        Calculate consistency regularization $\mathcal{L}_{con}$ on $\mathcal{B}_{dual}$
        Calculate pseudo-negative loss $\mathcal{L}_{od}^{neg}$ on $\mathcal{B}_{neg}$
        Calculate other losses $(\mathcal{L}_{od}^{em}, \mathcal{L}_{od}^{SOCR})$ on the full batch $\mathcal{B}$
        $\mathcal{L}_{OOD}^{unlab} \leftarrow \lambda_{em}\mathcal{L}_{od}^{em} + \lambda_{SOCR}\mathcal{L}_{od}^{SOCR} + \lambda_{neg}\mathcal{L}_{od}^{neg}$
        **if** $t \geq T_{align}$ **then**
            Calculate alignment loss $\mathcal{L}_{align}$ on the full batch $\mathcal{B}$
            $\mathcal{L}_{client} \leftarrow \lambda_{con}\mathcal{L}_{con} + \lambda_{align}\mathcal{L}_{align} + \lambda_{od}\mathcal{L}_{OOD}^{unlab}$
        **else**
            $\mathcal{L}_{client} \leftarrow \lambda_{con}\mathcal{L}_{con} + \lambda_{od}\mathcal{L}_{OOD}^{unlab}$
        Update $\theta_c$ by minimizing $\mathcal{L}_{client}$
    **return** *updated model $\theta_c$ and align loss $\mathcal{L}_{align}$*

---

---

**Algorithm 1:** OpenFL: (Server Procedure)

---

**Procedure** *ServerWarmup*($\theta_g, \theta_{EMA}$)

    **for** *epoch* = 1 *to* $E_w$ **do**

        Calculate supervised loss $\mathcal{L}_{server} \leftarrow \mathcal{L}_{ce} + \lambda_{od}\mathcal{L}_{od}^{OVA}$

        Update $\theta_g$ by minimizing $\mathcal{L}_{server}$ on data from $\mathcal{D}_l$

        $\theta_{EMA} \leftarrow \alpha \cdot \theta_{EMA} + (1 - \alpha) \cdot \theta_g$

    Update BN statistics of $\theta_g$ using $\mathcal{D}_l$

    Calculate initial pivots $\mu$ using features from the trained $\theta_g$

    **return** $\theta_g, \theta_{EMA}, \mu$

**Procedure** *ServerAggregation*($\{(\theta_c^{t+1}, \mathcal{L}_{align,c})\}$)

    **for** *each client* $c \in S_t$ *in parallel* **do**

        Calculate aggregation weights $\tilde{n}_c \leftarrow \frac{\kappa}{\mathcal{L}_{align,c}+\epsilon}$

    Update aggregated server model $\theta_g^{t+1} \leftarrow \frac{\sum_{c\in S_t} \tilde{n}_c \theta_c^{t+1}}{\sum_{c\in S_t} \tilde{n}_c}$

    **return** $\theta_g^{t+1}$

**Procedure** *ServerFineTune*($\theta_g, \theta_{EMA}, \mu$)

    Freeze BN layers of $\theta_g$

    **for** *epoch* = 1 *to* $E_s$ **do**

        Calculate supervised loss $\mathcal{L}_{server} \leftarrow \mathcal{L}_{ce} + \lambda_{od}\mathcal{L}_{od}^{OVA}$

        Update $\theta_g$ by minimizing $\mathcal{L}_{server}$ on data from $\mathcal{D}_l$

    $\theta_{EMA} \leftarrow \alpha \cdot \theta_{EMA} + (1 - \alpha) \cdot \theta_g$

    Update BN statistics of $\theta_g$ using $\mathcal{D}_l$

    $\mu \leftarrow$ **PivotUpdate**($\theta_{EMA}$)

    **return** $\theta_g, \theta_{EMA}, \mu$

**Procedure** *PivotUpdate*($\theta_{EMA}$)

    $h \leftarrow \mathcal{F}(\mathcal{D}_l; \theta_{EMA})$

    $z \leftarrow \text{Proj}(h)$

    **for** $k = 1$ *to* $K$ **do**

        $\mu_k = \text{mean}(z \mid y = k)$

    **return** $\mu$

---

