# OpenReview forum: "Global Pivots, Local Unknowns: Stable Federated Open-Set Semi-Supervised Learning"
_ICLR.cc/2026/Conference — Submitted to ICLR 2026_

### Official Review · Reviewer_9cdJ · 2025-10-27

**Soundness:** 2
**Presentation:** 3
**Contribution:** 2
**Rating:** 2
**Confidence:** 3

**Summary:**

1. The paper formalizes Federated Open-Set Semi-Supervised Learning (FOSSL) in a "labels-at-server" regime, where clients hold only unlabeled, non-IID, open-set data, posing challenges of pseudo-label brittleness and intensified heterogeneity from diverse unknown classes.
2. It proposes OpenFL, a server-guided framework that stabilizes training using three components: Round-wise EMA (R-EMA) for a stable server-side model, Pivot-guided Open-set Alignment to guide clients with stable class references, and Reliability-Aware Aggregation (RAA) to weight clients by update quality rather than data size.
3. OpenFL consistently improves both in-distribution (ID) accuracy and out-of-distribution (OOD) detection (AUROC) across CIFAR-10, CIFAR-100, and FashionMNIST, remaining stable where federated baselines fail.

While effective, the work is primarily compositional in nature, drawing from established techniques, and thus fails to provide sufficient novelty.

**Strengths:**

1. The open-set formulation is novel.
2. All the proposed components of the training method are effective and improve performance empirically.

**Weaknesses:**

1. Using a moving average at the server has been done before, both in a centralized iteration-based fashion and in a federated round-based setting[1,2].
2. Guiding aggregation weights by the loss has been done before [3,4], using specific losses is not sufficient grounds for novelty.
3. No theoretical guarantees are provided.
4. All experiments are conducted on small-scale computer vision datasets (FashionMNIST, CIFAR-10, CIFAR-100) using a relatively shallow backbone. These benchmarks do not adequately represent the challenges of modern deep learning. It is unclear if the proposed methods would remain effective when fine-tuning or pre-training large-scale foundation models.

[1] Zhang, et.al; "How Does Critical Batch Size Scale in Pre-training?"

[2] Zhou, et.al; "Understanding and Improving Model Averaging in Federated Learning on Heterogeneous Data"

[3] Li, et.al; "Fair Resource Allocation in Federated Learning"

[4] Li, et.al; "Tilted Empirical Risk Minimization"

**Questions:**

1. How does the computational complexity of your method scale as the model size increases, particularly with the embedding dimension and the number of classes?

---

> ### Author Response · Authors · 2025-12-03
>
> We thank the reviewer for recognizing the novelty of the open-set formulation and the empirical effectiveness of our components. Below, we address each concern in detail.
>
> **[W1 & W2] Novelty of R-EMA and Reliability-Aware Aggregation**
>
> Please refer to the Unified Clarification in our general response, where we provide a full discussion of FOSSL-specific novelty. Here, we respond specifically to the cited works ([1]–[4]).
>
> **(1) Why R-EMA is fundamentally different from EMA in prior work ([1], [2])**
>
> The cited EMA works serve entirely different purposes:
>
> - [1] Zhang et al. studies critical batch size and optimization dynamics in centralized pre-training; **it is unrelated to federated semi-supervised learning.**
>
> - [2] FedIMA applies EMA smoothing and broadcasts the smoothed model back to clients to mitigate supervised heterogeneity.
>
> **In contrast, R-EMA serves a fundamentally different role in FOSSL:**
>
> - **The EMA model is never broadcast to clients.**
> Unlike FedIMA which operates in a fully supervised setting, FOSSL relies on self-generated pseudo-labels. We explicitly avoid broadcasting the EMA model because its predictions become over-smoothed and unreliable under OOD contamination (a failure mode we diagnose in Sec. 3).
>
> - **R-EMA stabilizes global class-level pivots.**
> This is a unique requirement in FOSSL where:
>     - supervision exists only on the server,
>     - client updates are highly unstable due to OOD heterogeneity,
>     - and a consistent representational anchor is needed across clients.
>
> Thus, while EMA is a known tool, its purpose, integration, and behavior in *OpenFL* are specific to FOSSL and not comparable to prior work.
>
>
> **(2) Novelty of Reliability-Aware Aggregation (vs. [3], [4])**
>
> We respectfully argue that dismissing our aggregation mechanism as "using a specific loss" overlooks the core challenge of FOSSL: estimating client reliability without labels.
>
> - **Divergent Objectives (Convergence vs. Reliability):**
> Most weighting methods (e.g., [3]) treat loss as a proxy for difficulty, up-weighting high-loss clients to accelerate convergence or ensure fairness.
> In FOSSL, this logic is inverted: High loss typically indicates OOD contamination. Up-weighting them (as done in [3]) amplifies noise and accelerates collapse. Reliability-Aware Aggregation explicitly prioritizes reliability, not convergence speed.
>
> - **Blind Statistical Filter vs. Semantic Proxy:**
> We acknowledge that TERM [4] down-weights outliers based on loss magnitude. However, it acts as a blind statistical filter that suppresses all high-loss samples without distinguishing their causes. In FOSSL, high loss is ambiguous because it may reflect harmful OOD noise or valuable hard ID samples. TERM cannot differentiate between the two.
> *OpenFL* resolves this by using the **Pivot-Guided Alignment Loss as a semantic proxy**. Even when a hard ID sample has high loss, it still aligns structurally with its class pivot. This allows *OpenFL* to **down-weight clients whose updates are dominated by unreliable pseudo-labels caused by heavy OOD contamination**, while preserving contributions from clients that contain informative ID samples. Generic loss-based weighting cannot make this distinction.

---

> ### Author Response · Authors · 2025-12-03
>
> **[W3] Theoretical Guarantees**
>
> We acknowledge the importance of theoretical analysis. However, our focus is on empirically validating a solution for the novel FOSSL setting, where deriving tight convergence bounds remains an open challenge due to the complex interplay of OOD contamination, label scarcity, and highly non-IID client distributions.
>
> - **Intractability of Formal Bounds:**
> In such heterogeneous open-set environments, theoretical guarantees are currently intractable and often lead to very loose bounds with limited practical value. This limitation is not specific to our method; most influential works in FSSL and OSSL rely primarily on empirical validation rather than formal theory because the underlying problem structure is inherently difficult to characterize mathematically.
>
> - **Empirical Verification of Stability:**
> Instead of providing weak theoretical bounds, we rigorously evaluate *OpenFL*'s stability through empirical stress tests, including the sensitivity analysis in Fig. 4 and the failure-mode diagnosis in Sec. 3. These experiments collectively demonstrate that *OpenFL* reduces variance and stabilizes learning across heterogeneous client conditions.
>
> We view the theoretical characterization of pivot-guided alignment under open-set federated heterogeneity as a meaningful direction for future work.
>
>
> **[W4] Experimental Scale & Foundation Models**
>
> We respectfully clarify that our experimental choices are deliberate, adhering to both community benchmarks and the physical constraints of the target application.
>
> - **Adherence to Community Standards:**
> CIFAR-10/100 and Fashion-MNIST are the standard benchmarks used in the vast majority of recent OSSL and FSSL works (e.g., SemiFL, FedMatch). Using these datasets enables direct and fair comparison with prior methods under identical conditions.
>
> - **Realistic Constraints of Cross-Device FL:**
> FOSSL specifically targets cross-device settings (mobile/IoT), where clients are severely constrained in computation, memory, and communication bandwidth.
> Training or fine-tuning large foundation models on-device is impractical in this regime due to memory and communication limitations.
> Using lightweight backbones therefore reflects real-world deployment constraints and aligns with common practice in top-tier FL research, which prioritizes algorithmic robustness under realistic resource budgets.
>
> ---
>
> **[Q1] Computational Complexity & Scalability**
>
> *OpenFL* introduces negligible computational overhead, and its complexity is effectively decoupled from the model size.
>
> - **Decoupled Complexity via Projection:**
> Alignment loss and pivot updates are computed in a projected latent space ($D\_{proj}$), not in the full embedding dimension. Even if the backbone scales up, features are projected to a fixed and low dimension (e.g., 128), ensuring constant computational cost.
> - **Trivial Overhead:**
>     - Space: The server stores only $K$ pivot vectors of size $D\_{proj}$, which is negligible compared to model parameters.
>     - Time: Alignment calculations scale with $O(B \times K \times D\_{proj})$, where $B$ is the batch size.
> These operations are trivial relative to backbone forward/backward passes.
>
> Thus, *OpenFL* scales linearly and efficiently regardless of model size or number of classes.

---

### Official Review · Reviewer_wDvA · 2025-10-29

**Soundness:** 3
**Presentation:** 3
**Contribution:** 2
**Rating:** 4
**Confidence:** 4

**Summary:**

This paper attempts to address the problem of Federated Open-Set Semi-Supervised Learning (FOSSL) in labels-at-server setting. In this setting, a central server holds a small amount of labeled ID data, while clients possess only unlabeled data that contains both ID and OOD samples. The authors claim that existing methods fail due to pseudo-label brittleness and data heterogeneity. They propose OpenFL,which combines three main components: (1) R-EMA model on the server, (2) a pivot-guided alignment where clients align high-confidence samples to server-computed class prototypes, and (3) RAA scheme that weights clients based on the inverse of their alignment loss. The experiments show that their method achieves great ID accuracy and OOD detection.

**Strengths:**

1. The empirical study in Section 3, which demonstrates the failure modes of naively applying existing FSSL and OSSL methods to this setting, providing a clear motivation for the problem.
2. The paper is well-written and most parts are clearly explained.

**Weaknesses:**

1. OpenFL appears to be little more than a combination of existing, well-known ideas stitched together, such as exponential moving average (common strategy in SSL methods) and prototype learning. So in my opinion, the contribution is negligible.
2. The use of globally fixed thresholds for the dual-gate selection is sub-optimal. A good confidence score on a client with clean data might be a bad one on a client swamped with OOD samples.
3. The federated adaptations of centralized OSSL methods, particularly FedSCOMatch and FedProSub, perform exceptionally poorly, often leading to model collapse. Can the authors provide evidence that these are not strawman implementations? Please detail the specific adaptation strategies and hyperparameters used, and justify why you believe this represents a fair comparison.

**Questions:**

See in Weaknesses.

---

> ### Author Response · Authors · 2025-12-03
>
> We thank the reviewer for the insightful feedback and address each concern below.
>
> **[W1] Novelty Clarification.**
>
> Please refer to our unified clarification in the General Response to all reviewers.
> Even if individual components appear superficially similar to prior ideas, their behavior and feasibility fundamentally change under FOSSL due to the server-only labels, heterogeneous OOD contamination, and strong non-IID client distributions. Techniques such as EMA or prototype-based learning cannot be directly reused: in FOSSL, their standard forms either collapse or amplify errors (as shown in Sec. 3).
>
> *OpenFL*’s components were therefore designed specifically to correct failure modes that arise only in FOSSL, rather than assembled from existing methods.
>
>
> **[W2] Globally fixed thresholds.**
>
> We clarify that the use of global thresholds is a **deliberate design choice.**
>
> - **Risk of Local Adaptation:** Client-specific thresholds are fundamentally *unsafe in FOSSL*. OOD-heavy clients would artificially inflate their notion of "confidence" to meet local criteria, worsening the pseudo-label amplification issue diagnosed in Sec. 3.
>
> - **Role as a Safety Guard:** In *OpenFL*, the threshold functions only as a coarse safety filter. The primary robustness stems from the soft attraction/repulsion of the Pivot-Guided Alignment and R-EMA stabilization, which effectively handle boundary cases.
>
> - **Empirical Stability:** As shown in our sensitivity analysis, performance remains stable across a wide threshold range, confirming that global thresholds are both safe and sufficient in this setting.

---

> ### Author Response · Authors · 2025-12-03
>
> **[W3] Fairness of FedSCOMatch / FedProSub Adaptation**
>
> We respectfully clarify that the poor performance of FedSCOMatch and FedProSub is not due to implementation flaws. Both methods were faithfully adapted to the FOSSL setting, and their collapse arises from intrinsic structural incompatibilities between these centralized algorithms and the constraints of FOSSL.
>
> **1. Rigorous and Fair Adaptation Strategy**
>
> We employed a unified adaptation protocol for all centralized OSSL methods to ensure fairness:
>
> - **Component Distribution:** All components requiring labeled data are placed on the Server, and components requiring unlabeled data are placed on Clients.
>
> - **Pseudo-labeling:** We adopt a SemiFL-style design where pseudo-labels computed by the global server model are fixed during each client's local training.
>
> - **Hyperparameters:** All method-specific parameters were tuned following the original papers and validated through extensive experimentation.
>
> **Evidence of Sound Adaptation (The FedSSB Case):**
> Applying this exact same adaptation strategy to FedSSB results in a strong baseline performance in our experiments. This confirms that our adaptation framework is sound and that the collapse of SCOMatch and ProSub is due to their specific algorithmic assumptions, not implementation errors.
>
>
> **2. Method-Specific Details and Root Cause of Failure**
>
> Below, we detail the specific components implemented and explain why they are fundamentally incompatible with FOSSL.
>
> **(A) FedSCOMatch**
>
> - Implementation (Faithful to Original):
>
>     - Server: ID supervision loss ($\mathcal{L}\_{sup}^{id}$)
>
>     - Clients: OOD supervision using local OOD memory queues, along with Open-set ($\mathcal{L}\_{open}$) and Closed-set ($\mathcal{L}\_{close}$) self-training losses.
>
> - Why it Collapses (Structural Incompatibility):
> The original SCOMatch relies on a global OOD memory queue to learn diverse OOD features. In FL, privacy constraints prohibit data sharing, forcing each client to maintain an isolated OOD queue. This results in heterogeneous, fragmented OOD distributions, causing severe representation drift and feature collapse. This failure persists regardless of hyperparameter tuning.
>
> **(B) FedProSub**
>
> - Implementation (Faithful to Original):
>
>     - Server: Supervised ID loss + L2 regularization; updates Beta parameters ($\alpha\_{id}, \beta\_{id}$).
>
>     - Clients: Subspace loss, pseudo-labeling loss, self-supervision loss; update all Beta parameters ($\alpha\_{id}, \beta\_{id}, \alpha\_{ood}, \beta\_{ood}$).
>
>     - Aggregation: FedAvg followed by EMA smoothing.
>
> - Why it Collapses (Optimization Discrepancy):
> Centralized ProSub requires joint updates of parameters using both labeled and unlabeled samples within the same batch. In FOSSL, labeled data exists only on the server and unlabeled data only on clients, making this joint optimization mathematically impossible. Decoupling these updates causes the parameter estimates ($\alpha, \beta$) to diverge, leading to unstable optimization and model collapse.
>
> **The failure of FedSCOMatch and FedProSub stems from fundamental algorithmic mismatches**, specifically the reliance on shared memory queues and joint labeled-unlabeled optimization, **which cannot be realized under FOSSL constraints**. **This reinforces the necessity of *OpenFL*'s design**, which uses server-side pivots to bridge this gap without violating FL constraints.

---

### Official Review · Reviewer_bpFb · 2025-10-31

**Soundness:** 2
**Presentation:** 2
**Contribution:** 2
**Rating:** 4
**Confidence:** 4

**Summary:**

This work tackles the challenging Federated Open-Set Semi-Supervised Learning setting where labeled training data is uniquely located on the server. The authors proposed OpenFL, a novel framework that enables the server to control the federated training and guide clients to meaningfully contribute to the global model, even though they don’t possess any labelled training samples. OpenFL comprises a series of techniques such as round-wise exponential moving average, global pivots, and reliability-aware client weights aggregation.

**Strengths:**

- S1. The authors tackle a challenging and critical federated learning setting.
- S2. The authors propose a full end-to-end method to robustly train federated models where clients' training data is completely unlabelled.
- S3. The proposed method’s ML performance is evaluated across standard benchmarks in a meaningful setting.

**Weaknesses:**

- W1. Limited novelty – Many components of OpenFL (e.g., using a global pivot model or EMA updates) have been proposed previously (e.g., in recent federated semi-supervised methods like FedAnchor [1]). While OpenFL’s combination of these techniques in the FOSSL setting is useful, the approach feels incremental rather than introducing a fundamentally new concept.
- W2. Complex tuning – OpenFL introduces numerous new hyperparameters (e.g., for the EMA decay, pivot selection, client weighting). This added complexity could make the method hard to tune in practice, potentially limiting its real-world applicability. This concern is heightened by the fact that the experiments were on well-established benchmarks with presumably careful tuning; deploying OpenFL in the wild might be challenging without guidance on choosing these hyperparameters.
- W3. Limited evaluation scope – The experimental settings are not fully representative of challenging real-world federated scenarios. For instance, the paper evaluates on at most 20 clients with reasonably large local datasets, but does not test cases with a huge number of clients or with extremely scarce data per client. This omission leaves it unclear how OpenFL performs in more extreme or realistic federated conditions (e.g., hundreds of clients or clients with only a handful of samples).

[1] Xinchi Qiu, Yan Gao, Lorenzo Sani, Heng Pan, Wanru Zhao, Pedro PB Gusmao, Mina Alibeigi, Alex Iacob, and Nicholas D Lane. Fedanchor: Enhancing federated semi-supervised learning with label contrastive loss for unlabeled clients. arXiv preprint arXiv:2402.10191, 2024.

**Questions:**

- Q1. The abstract currently spends a lot of space on method details. Could the authors revise it to highlight the key challenges of the FOSSL setting more explicitly, rather than the implementation specifics of OpenFL?
- Q2. Can the authors clarify how abundant they assume the server training dataset is compared to the local client datasets? It is critical for setting the context of the applicability of the OpenFL method.
- Q3. Can the authors discuss more explicitly in the introduction what key challenges the components of OpenFL are meant to address?
- Q4. How would OpenFL perform in scenarios of extreme data scarcity? Consider two cases: (a) the server’s labeled dataset is very scarce relative to clients (e.g., only 1–10% the size of the total client data), and (b) each client’s local dataset is so small that a full batch can’t be formed without reusing data. Can the authors discuss how OpenFL would handle these situations?
- Q5. Despite being unpublished work, how do the authors think OpenFL compares to Fedanchor [1]? I would like to read their opinion comparing the two methods on: (a) general setting; (b) motivating examples; (c) basic working principle; (d) performance (if they have sufficient time to try to reproduce, but this point is not crucial).
- Q6. Can the authors add a fitting line in Figure 3 to help readability and quantify the correlation they claim?
- Q7. How many global pivots does OpenFL require to perform sufficiently well?
- Q8. Can the authors add to each table and figure reporting results from the ablation/sensitivity studies (tables 2 and 3, and figure 4) a horizontal line showing the performance of the best baseline method as well?
- Q9. Given that the authors used open-source software to implement and test OpenFL, will they make the code publicly available?

[1] Xinchi Qiu, Yan Gao, Lorenzo Sani, Heng Pan, Wanru Zhao, Pedro PB Gusmao, Mina Alibeigi, Alex Iacob, and Nicholas D Lane. Fedanchor: Enhancing federated semi-supervised learning with label contrastive loss for unlabeled clients. arXiv preprint arXiv:2402.10191, 2024.

---

> ### Author Response · Authors · 2025-12-03
>
> We thank the reviewer for the thoughtful comments.
> Several concerns appear to stem from a misunderstanding of our setting (e.g., the actual scale of clients and the role of pivots), and we first clarify these points and then respond to each item in detail.
>
>
> **[W1] Novelty Clarification**
>
> Please refer to our unified clarification in the General Response to all reviewers.
> Here, we specifically address the reviewer's comparison to FedAnchor, which operates under fundamentally different assumptions.
>
> FedAnchor assumes a **closed-set FSSL** setting in which all client samples belong to ID classes. Its anchors are sample-level embeddings of labeled server data, and unlabeled client features are pulled toward these anchors via a label-contrastive objective.
> This design tightly binds representation shaping to specific labeled examples, making it fragile under domain shift, intra-class variation, or heterogeneous client distributions. In addition, **communication cost grows with the number of anchors because all sample-level anchors must be transmitted.**
>
> In contrast, *OpenFL* is designed for the **open-set FSSL (FOSSL)** setting in which (1) clients hold unlabeled data that may contain substantial OOD samples, and (2) supervision resides only on the server.
> Instead of transmitting many sample-level anchors, we maintain **one class-level pivot per ID class**, updated via R-EMA model to reflect the broader federated distribution. These pivots act as stable, abstract global anchors that are robust to heterogeneity and OOD contamination.
> Alignment is performed through **soft attraction-repulsion**: ID-like samples are gently attracted, while uncertain or OOD-prone samples are mildly repelled.
> This mechanism allows *OpenFL* to leverage unlabeled data *even under OOD contamination*, a regime FedAnchor was not designed to handle.
>
> Thus, although both methods involve "alignment", their assumptions, alignment targets, communication behavior, and stability properties under open-set non-IID federated training are fundamentally different.
>
>
> **[W2] Complex Tuning**
>
> We appreciate the reviewer's concern regarding real-world usability. We clarify that *OpenFL* is intentionally designed to require minimal tuning, and part of the perceived complexity likely stems from incorrect assumptions about the framework.
>
> - **No Pivot Selection or Tunable Client Weighting.**
>
>     - **Pivot selection does not exist in *OpenFL*.** Each class has a single pivot defined as a server-side feature mean smoothed by R-EMA. No sampling or filtering is required.
>     - **Client weighting is parameter-free.** Weights are computed directly from the alignment loss, providing automatic reliability estimation.
>
> - **Mostly Standard Hyperparameters.**
> FixMatch confidence threshold, augmentations, OVA parameters, etc., are adopted as-is from standard SSL/OSSL literature (FixMatch, SSB), ensuring ease of reproduction.
>
> - **Only Two Method-Specific Hyperparameters.**
> *OpenFL* adds only the alignment-loss weight and OVA threshold. As shown in Fig. 4, performance is stable across broad ranges of both.
>
> - **R-EMA Decay Is Not Sensitive.**
> Although R-EMA includes a decay factor, experiments show that a standard range (0.90–0.95) consistently performs well across datasets and heterogeneity levels, acting simply as a smoothing constant rather than a sensitive hyperparameter.
>
> - **Robust Improvements Across All Settings.**
> While absolute metrics vary slightly with hyperparameter choices, *OpenFL* consistently outperforms all baselines under every tested configuration, indicating that gains arise from the framework design, not from finely tuned parameters.

---

> ### Author Response · Authors · 2025-12-03
>
> **[W3] Evaluation Scope & Real-World Applicability**
>
> We respectfully clarify that the reviewer's concern is based on an incorrect assumption.
> **Our experiments do not use 20 clients. We use 100 clients**, following standard cross-device FL protocols.
>
> - **Scale Clarification: 100 Clients (Cross-Device FL).**
> We clarify that our experiments involve **100 clients with a 10% participation rate** (10 per round). It presents a significantly more challenging optimization landscape than the cross-silo settings (e.g., 5–20 clients).
>
> - **Realistic Heterogeneity & Data Scarcity.**
> Our evaluation reflects a rigorous and realistic FOSSL scenario:
>
>     - Clients hold no labels, and the server has access to only 2-8% of the total samples for each ID class (relative to the full federated dataset).
>     - Client data may contain substantial OOD contamination, introducing severe noise and non-IID drift.
>
> - **Behavior under More Extreme Conditions.**
> Although simulating clients with only a handful of samples was not the main focus, the architecture of *OpenFL* naturally accommodates such regimes:
>
>     - R-EMA pivots provide a stable global reference even when local batches are tiny or unreliable.
>     - Instance-wise alignment does not depend on large local datasets and remains functional with sparse or imbalanced per-client data.
>
> Thus, our evaluation not only aligns with community standards but already represents a demanding and realistic FOSSL environment.
>
> ---
>
> **[Q1] Abstract revision.**
> We thank the reviewer for the helpful suggestion.
> The original abstract did mention the central challenges of FOSSL, but we agree that they can be highlighted more explicitly.
> Accordingly, we have revised the abstract to clearly state the novelty of introducing FOSSL, emphasize its fundamental challenges (pseudo-label brittleness and amplified heterogeneity under OOD contamination), and present *OpenFL* at a high level without unnecessary implementation details.
>
> **[Q2] Clarification of server training data assumptions.**
> We clarify that the server holds only 2–8% of the total samples per ID class, relative to the full federated dataset. This strictly falls into the label-scarce regime and is consistent with the realistic constraints discussed in our response to [W3].
>
> **[Q3] Purpose of *OpenFL* components.**
> We appreciate the suggestion. We have revised the introduction to explicitly map each component of *OpenFL* to the specific FOSSL failure mode it addresses.
>
> **[Q4] Extreme data scarcity.**
> As discussed in our response to [W3], *OpenFL* is explicitly designed for these conditions:
>
> - Server-side label scarcity: Our experiments already operate in the 2-8% regime, which directly matches the reviewer's "1-10%" scenario. *OpenFL* demonstrates strong performance here.
> - Tiny Client Batches: Since R-EMA pivots are maintained on the server, they provide a stable global reference that does not fluctuate with local batch sizes. Furthermore, our instance-level alignment operates on individual samples, ensuring stability even when clients cannot form full batches.
>
> **[Q5] Comparison with FedAnchor.**
> A detailed comparison covering the setting, working principles, and limitations of FedAnchor is provided in our response to [W1].
>
> **[Q6] Fitting line in Figure 3.**
> Thank you for the suggestion. We have added a regression line and Pearson correlation coefficient to Fig. 3 to improve readability and quantitatively confirm the correlation.
>
> **[Q7] Number of pivots.**
> As clarified in our response to W2-(1), *OpenFL* maintains exactly one global pivot per ID class.
>
> **[Q8] Adding a baseline reference line.**
> We appreciate the suggestion.
> However, we found that adding a single horizontal line may be misleading because no single baseline acts as the "best" performer across both Accuracy and AUROC simultaneously (due to the inherent trade-off in FOSSL). Adding multiple lines would create visual clutter.
> Instead, for reporting our main results, we use the default configuration (EMA decay = 0.9, $\lambda\_{align} = 1$, $\eta\_{ID} = 0.9$), which provides the most consistent and stable performance across all experiments.
>
> **[Q9] Code release.**
> We utilized the *Flower (Flwr) framework* as our infrastructure, and all algorithmic components were implemented by the authors. We can release our implementation publicly and will make the code available upon acceptance to support reproducibility.

---

### Official Review · Reviewer_pWQc · 2025-11-01

**Soundness:** 3
**Presentation:** 3
**Contribution:** 2
**Rating:** 4
**Confidence:** 3

**Summary:**

This paper addresses the Federated Open-Set Semi-Supervised Learning (FOSSL) problem, where the server has access to a small labeled dataset of in-distribution (ID) classes, while clients hold only unlabeled, non-IID data that may include out-of-distribution (OOD) samples.This paper proposes OpenFL with three components: Round-wise EMA (R-EMA): A round-wise exponential moving average model; Pivot-Guided Open-Set Alignment: Global pivots guide client-side alignment, attracting high-confidence ID samples while mildly repelling uncertain/OOD samples; Reliability-Aware Aggregation (RAA): Client contributions are weighted based on alignment loss.

**Strengths:**

1. Foucs on a practical and underexplored problem.1

2. Comprehensive evaluation. The study includes relevant baselines, such as SemiFL, FedFixMatch, and federated adaptations of centralized OSSL methods. The experiments cover diverse datasets, client partitioning schemes (IID and non-IID), and multiple challenging splits, providing a broad evaluation of the proposed method.

3. Good presentation and writing.

**Weaknesses:**

1. Technical Novelty

The proposed method combines widely adopted techniques, including EMA, prototype-based alignment, and loss reweighting mechanisms. Each of these components is well-established in related works. For example: EMA is a standard stabilization technique in many learning systems. Pivot-based alignment is a direct extension of prototype methods used in centralized contrastive learning and semi-supervised learning. Reliability-aware aggregation using alignment loss is conceptually similar to weighting schemes, e.g., importance sampling or quality-based aggregation.

The combination of these components is incremental and does not introduce a new \textbf{insight} or novel \textbf{technique}.

2. High Sensitivity to Hyperparameters. The method relies heavily on existing loss functions and their combinations (e.g., FixMatch consistency loss, OOD detection losses from OpenMatch, SSB, etc.). It introduce multiple hyperparameters, making the method parameter-sensitive and hard to be generalized. The sensitivity analysis in the experiments demonstrates that performance can vary significantly depending on these choices.

3. Limited Applicability to Real-World Federated Settings

Server-side pivots are computed from a small labeled dataset, limiting the scalability of real-world application. If possible, please use large-scale dataset.

**Questions:**

Beyond the above concerns on Weaknesses, please answer:

1. Novelty Clarification

2. Dataset Scalability

3. Sensitivity Analysis of Hyperparameters Across Different Settings and Datasets

The hyperparameters in this method are highly sensitive across different datasets and settings.

---

> ### Author Response · Authors · 2025-12-03
>
> We thank the reviewer for the detailed and constructive feedback, and we address each point below.
>
>
> **[Q1, W1] Novelty Clarification**
>
> Please refer to our unified clarification in the General Response to all reviewers.
> FOSSL introduces a fundamentally different problem structure (server-only labels, unlabeled OOD-contaminated clients, and severe non-IID distributions), which leads to failure modes not previously identified in FL, FSSL, or OSSL. *OpenFL* is therefore not a combination of existing techniques, but **a framework whose components are specifically designed to address these FOSSL-specific failure modes.**
>
>
> **[Q2, W3] Dataset Scalability & Applicability**
>
> We respectfully clarify that our experimental design adheres to community standards while reflecting the constraints intrinsic to FOSSL.
>
> - **Adherence to Standard Protocols:**
> CIFAR-10/100 are the primary benchmarks used in the vast majority of OSSL and FSSL literature (e.g., SemiFL, FedMatch, SSB, SCOMatch, ProSub).
> We additionally evaluate on Fashion-MNIST, a widely adopted benchmark in FL alongside SVHN.
> These choices ensure fair and direct comparison with prior work.
>
> - **Label Scarcity is Intentional:**
> A defining challenge in FOSSL is the scarcity of server-side labels.
> Our experiments use a rigorous low-label regime (2–8% labeled samples), reflecting realistic scenarios where annotation is costly.
> Strong performance under this constraint demonstrates *OpenFL*’s suitability for practical, low-label deployments.
>
> - **Scalability & Pivot Robustness:**
> Our results show consistent improvement as the server label budget increases (CIFAR-10), indicating natural scalability. Moreover, R-EMA continuously refines the global pivots throughout training so that they gradually reflect the broader federated data distribution, helping maintain stable performance even under heterogeneous client compositions.
> Taken together, these design choices align with community standards and confirm that *OpenFL* scales reliably in realistic FOSSL environments.
>
>
> **[Q3, W2] Sensitivity Analysis**
>
> We respectfully clarify that *OpenFL* is designed to minimize tuning overhead and remain robust across different conditions.
>
> - **Adoption of Standard Configurations:**
> Most hyperparameters in *OpenFL* (e.g., FixMatch confidence threshold, augmentation strategies, OVA loss parameters) directly follow established SSL and OSSL literature such as FixMatch and SSB. We adopt these without modification to ensure fairness and reproducibility.
>
> - **Minimal Additional Tuning:**
> *OpenFL* introduces only two method-specific hyperparameters: the alignment loss weight and the OVA threshold. As shown in our sensitivity analysis (Figure 4), performance is stable across a broad range of values, indicating that *OpenFL* does not require fine-grained tuning.
>
> - **Consistent Relative Gains:**
> While absolute performance naturally varies with hyperparameter choices, *OpenFL* consistently outperforms all baselines across datasets and settings. This demonstrates that the performance gains arise from the inherent robustness of our mechanisms rather than from favorable hyperparameter tuning.
>
> Therefore, *OpenFL* does not exhibit unusual sensitivity or tuning burden beyond what is typical in SSL/OSSL. The framework is designed with practicality and robustness in mind, even under the challenging constraints of FOSSL.

---

### Author Response · Authors · 2025-12-03
**Unified Clarification on Novelty and Core Contributions**

We respectfully thank all reviewers for their thoughtful and constructive feedback.
As several reviewers raised questions regarding novelty, we provide a unified clarification here to contextualize our contributions.

**The contributions of *OpenFL* arise from the fundamentally *new* Federated Open-Set Semi-supervised Learning (FOSSL) setting introduced in this work.**
In this configuration, (i) labels exist only on the server, (ii) clients hold purely unlabeled and OOD-contaminated data, and (iii) training occurs under severe non-IID distributions. **No existing FSSL or OSSL method was designed for this scenario, and they fail to operate reliably under these constraints.**

**Our work is therefore driven by the need to identify and address the unique failure modes that emerge only in FOSSL,** rather than applying existing techniques directly.
*OpenFL* is built on two core contributions: (1) the first systematic diagnosis of failure modes inherent to FOSSL, and (2) FOSSL-specific mechanisms developed to remain stable under these challenges.

---

**(1) Previously unreported failure modes unique to FOSSL**

The configuration of labels-at-server + unlabeled open-set clients + non-IID distributions has been underexplored in FL. Our systematic empirical study (Sec. 3) reveals four fundamental failure modes that cause existing pipelines to collapse:

- **Amplification of incorrect pseudo-labels** caused by distributed OOD contamination, leading to rapid collapse in FSSL methods (e.g., SemiFL).
- **BatchNorm corruption**, where client-side OOD causes all BN variants except StaticBN to fail.
- **Federated OSSL divergence**, where strong centralized methods (SCOMatch, ProSub) break down due to disjoint supervision, heterogeneous OOD exposure, and aggregation drift.
- **The dual role of OOD**, which can benefit both inlier classification and OOD detection, but destabilizes training through sharp round-to-round fluctuations.

To the best of our knowledge, these dynamics have not been reported in prior FL, FSSL, or OSSL works, and **represent the first diagnostic study of the FOSSL setting.**

---

**(2) FOSSL-specific mechanisms enabling stable training**

Guided by these findings, we introduce **mechanisms purpose-built for FOSSL.**
Crucially, **conventional ideas and their standard variants, while effective in prior SSL or FL settings, fail or become substantially suboptimal under this setting.**

- **R-EMA (vs. Standard EMA):**
In centralized SSL, EMA is used solely as a smoothed evaluation model. Works like FedIMA [1] broadcast smoothed models to clients, but such use is unsuitable for FOSSL because EMA predictions become over-smoothed and unreliable for pseudo-labeling under OOD contamination.
We therefore keep the EMA model strictly on the server and use it only to stabilize global pivots. R-EMA is a mechanism specifically tailored to maintain a stable representational anchor in FOSSL.
- **Pivot-Guided Alignment (vs. Prototype Learning):**
This mechanism addresses the server–client disjoint supervision gap by using server-derived pivots as global representational anchors, allowing the server’s labeled signal to guide client-side representation learning despite the lack of client labels.
Conventional prototype methods rely on hard pseudo-labels, which become unreliable under OOD contamination and lead to amplification of incorrect pseudo-labels during local training. In contrast, Pivot-Guided Alignment adopts a soft attraction–repulsion strategy that aligns confident ID samples toward pivots while gently repelling uncertain or OOD-prone samples, enabling effective use of all unlabeled data without propagating pseudo-label errors.
- **Reliability-Aware Aggregation (vs. Importance Sampling):**
This is not a generic loss-based weighting scheme. Conventional weighting (e.g., up-weighting high-loss clients for fairness) would be unsafe in FOSSL because OOD-heavy clients naturally incur higher losses and would therefore receive more weight.
Our approach intentionally inverts this logic: the pivot-guided alignment loss provides a reliability signal that down-weights OOD-heavy clients and preserves ID representations. Thus, it functions as a defense mechanism against OOD drift, rather than as a convergence-oriented reweighting strategy.

**In summary,** *OpenFL* is not a simple combination of existing ideas.
It is a synergistic framework whose components are purpose-built to overcome failure modes that surface only in the newly introduced FOSSL setting.
These contributions are enabled by diagnosing what breaks under FOSSL and designing mechanisms that remain stable in the presence of unlabeled, OOD-contaminated, and highly non-IID client data.

---

Reference:
[1] Zhou et al., Understanding and Improving Model Averaging in Federated Learning on Heterogeneous Data

---

### Meta-Review · Area_Chair_b4XZ · 2026-01-02

**Summary:**

This paper formalizes the problem of Federated Open-Set Semi-Supervised Learning (FOSSL) under a labels-at-server setting, where clients possess only unlabeled, non-IID data with open-set classes, leading to pseudo-label instability and exacerbated client heterogeneity due to diverse unknown categories. To address these challenges, the authors propose OpenFL, a server-guided federated framework composed of three mechanisms: Round-wise EMA (R-EMA) to stabilize the server model, Pivot-guided Open-set Alignment to provide clients with consistent class-level references, and Reliability-Aware Aggregation (RAA) to weight client updates based on estimated update quality rather than data volume. Empirical results on CIFAR-10, CIFAR-100, and FashionMNIST demonstrate consistent improvements in both in-distribution accuracy and out-of-distribution detection performance (AUROC), particularly in scenarios where existing federated baselines become unstable. Despite these empirical gains, the proposed approach is largely compositional, relying on well-established techniques such as EMA stabilization, reference-based alignment, and weighted aggregation, and does not introduce fundamentally new algorithmic insights or theoretical understanding of FOSSL. As a result, the level of novelty appears insufficient, and the paper leans toward rejection.

**Reviewer Scores:**

No

---

### Decision · Program_Chairs · 2026-01-26

Reject